

# Control of wind speed and contact angle on submicron particulate matter sampling

Bokun Sun[1,2], Ziyang Wang[1], Jiayun Huang[1], Yumeng Li[1], James R. Cooper[2], Lei Han[1] and Bailiang Li[1]

[1] Xi'an Jiaotong-Liverpool University, Suzhou, China
[2] University of Liverpool, Liverpool, United Kingdom

## ABSTRACT

While particulate matter (PM) instruments are widely used for air quality monitoring and policy development, there is limited research on how wind speed ($U_0$) and contact angle ($\theta$) affect the measurement accuracy of submicron PM, or particles with their diameters $\leq$ one $\mu$m ($PM_1$). This study addresses this gap by employing a wind tunnel experiment with a common sampling system featuring a customized thick-walled cylindrical metal inlet. The results reveal that wind-induced aerosol losses can reach up to 9%, and the sampling efficiency has a negative linear relationship with $U_0$ and a third-order polynomial relationship with $\cos(\theta)$. This model demonstrates a significant discrepancy with classic models in predicting sampling efficiency, which indicates the inapplicability of classic sampling efficiency models to submicron aerosols. The findings of this study can help correct wind-induced errors, improve sampling protocols, and develop new predictive models, which have implications for improving hazardous air quality warning systems and safeguarding public well-being.

Corresponding author
Bailiang Li, bailiang.li@xjtlu.edu.cn

## INTRODUCTION

Long-term exposure to particulate matter (PM) can lead to respiratory ailments and, in severe cases, cardiovascular diseases (*Schwartz, Dockery & Neas, 1996*; *Yanosky, Williams & MacIntosh, 2002*). In particular, very fine PMs, such as $PM_{2.5}$ and $PM_1$ (here, subscript stands for the maximum particle diameter in $\mu$m), are associated with the most serious incidence and mortality of respiratory diseases (*Brown et al., 2001*; *Chuang et al., 2005*; *Yang et al., 2019*). Therefore, PM has been widely considered a major air pollutant.

Timely, accurate, and efficient aerosol sampling is critically important across numerous fields. For example, in occupational and environmental health, accurate aerosol measurement in real-time is essential for assessing health risks associated with airborne particles (*Hinds & Zhu, 2022*; *Rastmanesh et al., 2024*) and issuing public health warnings during acute air pollution events or enabling prompt interventions in workplaces to mitigate exposure to hazardous aerosols (*Luo et al., 2021*; *Qiu et al., 2022*). In atmospheric sciences, precise aerosol measurements with high spatiotemporal resolution are vital for

understanding complex atmospheric processes and improving the accuracy of air quality and climate models (*Carmichael et al., 2008*; *Li et al., 2024*; *Wieprecht et al., 2005*).

Traditionally, PM concentration is measured by weighing a filter before and after the aerosol sampling with a known flow rate (*Whalley & Zandi, 2016*). Although it is commonly adopted as a reference to assess other techniques (*EPA, 2017*), this method cannot provide real-time measurements. A number of instruments have been deployed to measure real-time PM gravimetric concentrations. Several common instruments, such as the Synchronized Hybrid Ambient Real-time Particulate (*e.g.*, Thermo Fisher Scientific SHARP 5030) (*Su et al., 2018*), Tapered Element Oscillating Microbalance (*e.g.*, Thermo Fisher Scientific TEOM 1400ab) (*Winkel et al., 2015*), and Beta Attenuation Monitor (*e.g.*, Met One BAM 1020) (*Gobeli, Schloesser & Pottberg, 2008*), though bulky, can directly measure the gravimetric PM concentrations. Conversely, most portable devices, such as TSI DustTrak 8530 (*Kushwaha et al., 2022*), Thermo Fisher Scientific PDM3600 and PDM3700 (*Tuchman et al., 2024*), and Shinyei PPD42NS (*Liu et al., 2017*), are based on traditional single-angle light scattering technology. Therefore, they can only indirectly derive the gravimetric PM concentration based on known PM density. Recent advancements in sampler development emphasize the integration of multiple analytical functions and the adoption of novel sensing technologies. For instance, a newly designed sampler combines gravimetric particulate matter (PM) sampling with continuous black carbon monitoring *via* optical backscatter techniques, allowing for simultaneous assessment of mass concentration and compositional properties. Emerging sensing technologies, such as static light scattering (*Ye et al., 2022*), resonant silicon microcantilever (*Xu & Peiner, 2023*), and multi-angle light scattering (*Huang et al., 2025*; *Zhao, 2023*), have been incorporated into PM measurement systems. However, irrespective of their sensing technologies, these instruments often experience aerosol loss or gain during air intake through inlets and tubes (*Brockmann, McMurry & Liu, 1982*), which can lead to variations in sampling efficiency and inaccurate measurement of PM concentration.

Sampling efficiency, $\eta_{sampling}$, is defined as the fraction of aerosols from the environment reaching the measurement location. This efficiency is given by the product of the aspiration efficiency $\eta_{aspiration}$, transmission efficiency $\eta_{transmission}$, and transport efficiency $\eta_{transport}$ (*Baron & Willeke, 2001*; *Brockmann, McMurry & Liu, 1982*; *Von der Weiden, Drewnick & Borrmann, 2009*):

$$\eta_{sampling} = \eta_{aspiration} \cdot \eta_{transmission} \cdot \eta_{transport} \tag{1}$$

Here, $\eta_{aspiration}$ represents the fraction of aerosol concentration in the ambient environment that is aspirated into the inlet, $\eta_{transmission}$ is defined as the fraction of aspirated aerosol concentration that successfully traverses the tubing system, and $\eta_{transport}$ is the ratio of the aerosol concentration at the beginning of the tubing system to that at the end. The first two efficiencies are mainly due to the aerosol loss occurring at the inlet. Therefore, the product of $\eta_{aspiration}$ and $\eta_{transmission}$ can also be termed as the inlet efficiency $\eta_{inlet}$, *i.e.*,

$$\eta_{inlet} = \eta_{aspiration} \cdot \eta_{transmission}. \tag{2}$$

Then,

$$\eta_{sampling} = \eta_{inlet} \cdot \eta_{transport}. \tag{3}$$

Generally, the inlet efficiency $\eta_{inlet}$ (comprising both $\eta_{aspiration}$ and $\eta_{transmission}$) is influenced by several factors, such as aerosol characteristics (*e.g.*, particle size, shape, and phase), sampler inlet and internal design, internal flow rate, and ambient environmental conditions (*e.g.*, temperature, relative humidity, and wind speed and direction). These factors often interact with each other, and the effect of certain factors may be completely opposite when other factors are changed.

Particle size is arguably the most critical particle property influencing sampling (*Fennelly, 2020*). For larger particles, their trajectories are more affected by changes in airflow direction near the inlet due to significant inertial and gravitational forces, while smaller particles can follow streamlines more effectively. Therefore, smaller particles normally have higher inlet efficiency (*Willeke, Lin & Grinshpun, 1998*). This principle has also been widely used for particle size separation, *e.g.*, cascade impactors (*Le & Tsai, 2021*) and cyclones (*Sigaev et al., 2006*).

The shape, phase, and chemical composition of aerosol particles significantly influence their aerodynamic behavior and, consequently, their $\eta_{inlet}$. For example, fibers with different orientations settle at significantly different speeds (*Kulkarni et al., 2011*). Solid particles may rebound when impacting the inlet wall, while liquid particles usually rest on the wall without further resuspension, which normally leads to higher inlet efficiency for solid particles (*Koehler et al., 2012*). Particles containing hygroscopic or volatile components may significantly enlarge or reduce their sizes under different air humidity and temperature conditions (*Meyer et al., 2009*; *Yu et al., 2025*), resulting in the variations of $\eta_{inlet}$.

The inlet's shape, dimension, and orientation define the airflow patterns in its immediate vicinity and control how particles are drawn from the ambient environment into the sampler (*Brockmann, 2011*). Thin-walled cylindrical inlets are probably the most commonly used and studied (*Belyaev & Levin, 1972*). According to *Belyaev & Levin (1974)*, when the ratio of outer diameter to inner diameter of the nozzle is smaller than 1.1, or the wall thickness is very small, the inlet can be treated as "thin-walled". Under such conditions, the aerosol loss at the leading edge can be negligible. Conversely, if the wall is too thick, the aspiration efficiency can drop significantly. Some inlets have a blunt design (*e.g.*, thick-walled cylindrical, disklike, and spherical) that might distort the upstream flow field, therefore affecting aspiration (*Chung & Dunn-Rankin, 1997*; *Chung & Ogden, 1986*; *Vincent, 1987*). The inlet orientation with respect to gravity is a crucial factor, especially for samplers targeting larger PMs, due to the significant influence of gravitational forces on these particles (*Hangal & Willeke, 1992*; *Okazaki, Wiener & Willeke, 1987b*; *Tufto & Willeke, 1982*).

Wind speed ($U_0$) plays a very important role in controlling inlet efficiency. The inlet dimension and internal flow rate control the flow speed in the inlet ($U$), and the ratio ($R_u$) of $U_0$ to $U$, or $R_u = U_0/U$, defines the isokinetic condition, which has a strong impact on aspiration efficiencies (c.f. Table 1). Ideally, for thin-walled samplers, under isokinetic conditions when $R_u = 1$, $\eta_{aspiration} = 1$, while large sampling errors can occur

**Table 1  Summary of previous experimental studies on aspiration and transmission efficiencies.**

| Study | Efficiency type | $D_p$(μm) | Particle type | Inlet Type | θ | St | $R_u$ |
|---|---|---|---|---|---|---|---|
| *Belyaev & Levin (1972)* | Aspiration | Not specified | Willow pollen and Lycopodium spores | Thin-walled and thick-walled | 0° | 0.18–2.03 | 0.17–5.6 |
| *Durham & Lundgren (1980)* | Aspiration | 1.0–11.1 | 90% uranine and 10% methylene particles; | Thin-walled | 0°–90° | 0.007–3 | 0.5–2.3 |
| | | 19.9 | Ragweed pollen | | | | |
| *Davies & Subari (1982)* | Aspiration and transmission | 14–30 | Di-2-cthylhexyl sebacate droplets | Thin-walled | 0°, 90° | 0.06–6.89 | 0.06–1 |
| *Tufto & Willeke (1982)* | Inlet | 2.5–20 | Uranine-tagged oleic acid droplets | Thin-walled | 0°–90° | 0.02–3.0 | 0.2–5.0 |
| *Vincent et al. (1986)* | Aspiration | 6-34 | Alumina dusts | Thin-walled | 0°–180° | 0.01–0.7[*] | 0.67-2.00 |
| *Lipatov et al. (1986)* | Aspiration | 34–80 | Water droplets | Thin-walled | 0° | 0.2–2.1 | 1.67–0.03 |
| *Chung & Ogden (1986)* | Aspiration | 3.4–12.5 | Diisooctyl phthalate powders | Disc-shaped blunt | 0° | 0.03–4.0 | Not specified |
| *Lipatov et al. (1988)* | Aspiration | 31 | Lycopodium spores | Thin-walled | 0° | Not specified | 0.1-1.0 |
| *Okazaki, Wiener & Willeke (1987a)* | Aspiration and transmission | 5–40 | Oleic acid droplets | Thin-walled | 0° | 0.01–10 | 0.25–8 |
| *Okazaki, Wiener & Willeke (1987b)* | Aspiration and transmission | 5–40 | Oleic acid droplets | Thin-walled | 0°–90° | 0.02–2 | 0.5–4 |
| *Wiener, Okazaki & Willeke (1988)* | Aspiration and transmission | 5–40 | Oleic acid droplets | Thin-walled | 0° | 0.1–5.0 | 0.5–2 |
| *Hangal & Willeke (1992)* | Aspiration and transmission | 10–40 | Oleic acid droplets | Thin-walled | 0°–20° | 0.13–0.52 | 0.5–2 |
| *Sreenath, Ramachandran & Vincent (2001)* | Aspiration and transmission | 1–37 | Glass beads | Thin-walled | 0°–90° | 0.002–2.0[*] | 0.38–3.8 |
| *Sreenath, Ramachandran & Vincent (2002)* | Aspiration and transmission | 1–37 | Glass beads | Blunt (spherical) | 0°–180° | 0.002–1.0[*] | 0.38–3.8 |
| *Paik & Vincent (2002a)* | Aspiration | 13-89.5 | Alumina powders | Thin-walled | 0° | 0.05–3.7 | 0.5–50 |
| *Paik & Vincent (2002b)* | Aspiration | 13-89.5 | Alumina powders | Disc-shaped blunt | 0° | 0.05–3.7 | 0.5–25 |
| *Li & Lundgren (2002)* | Aspiration | 5–681.6 | Ammonium-fluorescein powders; Uranine powders | Thin-walled & Blunt | 0°, 90°, 180° | Not specified | Not specified ($U_0 = 0.55$ or 1 m/s) |
| *Su & Vincent (2002)* | Aspiration | 40–70 | Olive oil droplets | Thin-walled | NA | 0.3–63[*] | 0 |
| *Su & Vincent (2003)* | Aspiration | 40–70 | Olive oil droplets | Blunt (spherical) | NA | 0.3–46.9 | 0 |
| *Brixey, Evans & Vincent (2005)* | Aspiration | 0.5–20 | Glass beads | Thin-walled | 90° | 0.0001–0.2[*] | 2 |

**Notes.**
   [*]Symbol indicates the numbers were estimated from figures, and $D_p$, θ, St, $R_u$, and $U_0$ are particle size, wind contact angle, Stokes number, wind to inlet flow speed ratio, and wind speed, respectively.

during anisokinetic conditions (*Brockmann, 2011*; *Liu, 2013*; *Rader & Marple, 1988*). A number of semi-empirical models have been proposed to correlate different components of sampling efficiency with $R_u$ and Stokes number $St$. For example, *Belyaev & Levin (1974)* proposed the following correlation with aspiration efficiency for typical thin-walled inlets that was later verified for the ranges of $0.2 < R_u < 5$ and $0.005 < St < 10$ as,

$$\eta_{aspiration} = 1 + (R_u - 1)(1 - \frac{1}{1+G}) \tag{4}$$

where $G = (2 + 0.617\text{R})\,St$. According to *Baron & Willeke (2001)*, $St = D_p^2 c \rho_p U_0 / (18 \mu D)$, where $D_p$ and $\rho_p$ are particle diameter and density, respectively, $\mu$ is the air dynamic viscosity, $D$ is the inner diameter of the nozzle, and $c$ is the Cunningham slip correction factor (*Allen & Raabe, 1985*) and can be evaluated using $c = 1 + Kn(1.142 + 0.558 \exp(-0.999/Kn))$. Here Knudsen number $Kn = 2\lambda/d$, and $\lambda$ is the gas molecular mean free path, which is a constant at a given atmospheric pressure and temperature (*Tsalikis, Mavrantzas & Pratsinis, 2024*). Other numerical studies (*Liu, Zhang & Kuehn, 1989*; *Zhang & Liu, 1989*) also proposed some models to estimate $\eta_{aspiration}$, and the simulation results are in good agreement with those from Belyaev & Levin's model. However, a more recent empirical study (*Su & Vincent, 2002*) shows that Belyaev & Levin's model is only good for $0.03 < R_u < 6$, and an improved $G$ estimation to cover a wider $R_u$ range of $0.03 < R_u < 50$: $G = 2 + (0.62/R_u) - R_u^{0.1}$ for $0.04 < St < 3.68$. Conversely, Vincent and his colleagues, after considering the effect of diverging streamlines around the blunt body and converging streamlines entering the inlet (*Tsai & Vincent, 1993*; *Tsai et al., 1995*; *Vincent, 1987*; *Vincent, 1989*), proposed a model to estimate $\eta_{aspiration}$ for thick-walled or blunt samplers,

$$\eta_{aspiration} = \left[ 1 + \alpha_1 \left( \frac{S^2}{D_b^2 \varphi} - 1 \right) \right] \left[ 1 + \alpha_2 \left( \frac{D^2}{S^2} - 1 \right) \right] \tag{5}$$

$$\alpha_1 = 1 - \frac{1}{1 + g_1} \tag{6}$$

$$\alpha_2 = 1 - \frac{1}{1 + g_2} \tag{7}$$

where $g_1$ and $g_2$ are empirical coefficients, $\varphi = r^2/R$, $r = D/D_b$, and $D_b$ is the sampler body diameter. The term $S$ is a length scale defining a 'stagnation' region over the sampler surface, which can be estimated using,

$$S = B\varphi^{1/3} D_b \tag{8}$$

Here, $B$ is a dimensionless term defining the bluntness of the sampler. *Chung & Ogden (1986)*, based on a limited range of $St$ (*i.e.*, $0.03 < St < 4$) and an unknown range of $R_u$, proposed $g_1 = 0.25$ and $g_2 = 6.0$ for a disklike sampler. *Vincent (1987)* further proposed a number of estimates for various types of blunt samplers, *e.g.*, $B = 1$, $g_1 = 0.25$, and $g_2 = 6.0$ for disklike samplers; $B = 0.6$, $g_1 = 0.15$, and $g_2 = 1.2$ for cylindrical thick-walled samplers; and $B = 0$, $g_1 = 2.1$, and $g_2 = 0$ for thin-walled samplers. However, the above aspiration efficiency models are not applicable for calm air conditions when $R_u = 0$. *Su & Vincent (2004)*; *Su & Vincent (2005)*, based on a number of empirical studies (*Su & Vincent, 2002*; *Su & Vincent, 2003*), proposed a general model to estimate aspiration efficiency for both thin-walled and blunt samplers in calm air:

$$\eta_{aspiration,calm} = 1 - 0.8 \left( 4 St_c R_c^{\frac{3}{2}} \right) + 0.08 \left( 4 St_c R_c^{\frac{3}{2}} \right)^2 - \beta_1 \left[ 0.5 R_c^{\frac{1}{2}} + R_c \left( 1/r^2 - 1 \right) \right] -$$

$$\beta_2 \left[ 0.12 R_c^{-0.4} \left( e^{-p} - e^{-q} \right) - R_c^{\frac{3}{2}} \left( 1/r^{\frac{1}{2}} - 1 \right) \right] \tag{9}$$

where $\eta_{aspiration,calm}$ is the aspiration efficiency in calm air, $\beta_1 = 0.8$, $\beta_1 = 0.2$ for horizontal sampling, and $\beta_1 = 0$ and $\beta_1 = 1$ for facing up sampling, $\beta_1 = 1$ and $\beta_1 = 0$ for facing down sampling, $St_c = D_p^2 c \rho_p U / (18 \mu D)$, $p = 2.2 R_c^{1.3} St_c$, $q = 75 R_c^{1.7} St_c$, and $R_c = D_p^2 c \rho_p g / (18 \mu U)$. This equation is valid for $0.001 < R_c < 0.1$, $0.5 < R_c < 50$, $4 St_c R_c^{3/2} \ll 1$, and $0.02 < r \leq 1$.

Wind speed can also affect the $\eta_{transmission}$ (c.f. Table 1), which is also controlled by gravitational and inertial losses as,

$$\eta_{transmission} = \eta_{tg} \eta_{ti} \tag{10}$$

where $\eta_{tg}$ and $\eta_{ti}$ are the gravitational and inertial transmission efficiencies, respectively. Gravitational loss occurs due to particle settling, and $\eta_{tg}$ can be modelled using (*Hangal & Willeke, 1990b*),

$$\eta_{tg} = \exp[-4.7 K^{0.75}] \tag{11}$$

where $K = Z^{1/2} St^{1/2} Re^{-1/4}$, $Z = D_p^2 c g \rho_p L / (18 \mu D U)$, $L$ is the inlet length, Reynolds number $Re = UD/\upsilon$, and $\upsilon$ is the air kinematic viscosity. Conversely, based on a number of studies (*Grinshpun, Willeke & Kalatoor, 1993*; *Okazaki, Wiener & Willeke, 1987a*; *Okazaki, Wiener & Willeke, 1987b*; *Sreenath, Ramachandran & Vincent, 2001*; *Tufto & Willeke, 1982*), two scenarios can lead to inertial loss. First, when $R_u > 1$, fast-moving particles can impact the inner wall once they enter the slower flows in the inlet boundary layer and cause the inertial loss, which can be quantified using the following equation for the ranges of $1 < R_u < 10$ and $0.01 < St < 100$ (*Liu, Zhang & Kuehn, 1989*),

$$\eta_{ti} = \frac{1 + (R_u - 1)/(1 + 2.66/St^{2/3})}{1 + (R_u - 1)/(1 + 0.418/St)}. \tag{12}$$

Second, when $R_u < 1$, *vena contracta* starts to emerge, and some inertial particles can be trapped in these regions and impact the inner wall (*McCabe, Smith & Harriott, 1993*). This effect can be expressed using the following correlation for $0.25 < R_u < 1$ and $0.01 < St < 100$ (*Hangal & Willeke, 1990a*):

$$\eta_{ti} = \exp(-75 I_v^2) \tag{13}$$

where $I_v = 0.09 St^{0.3}(-1 + 1/R_u)^{0.3}$.

Besides wind speed, wind direction or contact angle ($\theta$) can also influence both $\eta_{aspiration}$ and $\eta_{transmission}$ (c.f. Table 1). For the same wind speed with varying wind directions or contact angles, the curvature of the streamlines can also change dramatically (c.f. Figs. 1A–1E), leading to a significant difference in aerosol deposition on the outer wall. In addition, aerosol loss due to gravitational settling, *vena contracta* trapping, and inertial impaction loss at the inner wall may also vary with contact angle. Several studies indicate that the inlet efficiency for thin-walled inlets decreases as the contact angle increases from 0° to 90° in a horizontal plane due to increasing inertial impaction (*Davies & Subari, 1982*; *Durham & Lundgren, 1980*; *Li & Lundgren, 2002*; *Sreenath, Ramachandran & Vincent, 2001*; *Vincent et al., 1986*), and upward sampling has lower inlet efficiency than downward sampling
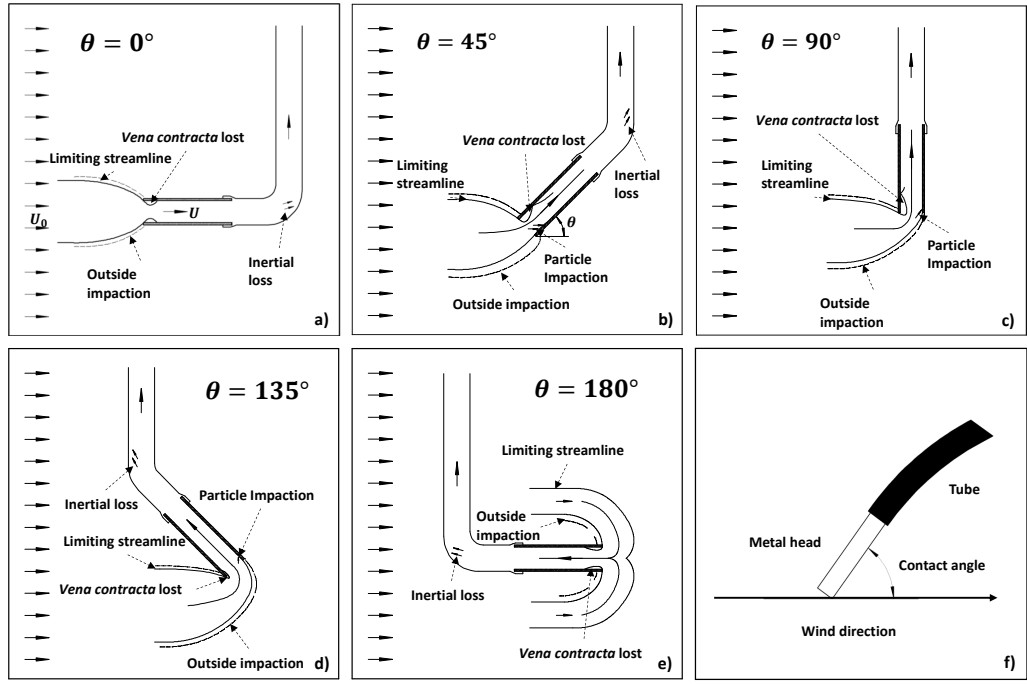

**Figure 1** Flow structure and aerosol processes for different contact angles (θ): (A) 0°, (B) 45°, (C) 90°, (D) 135°, and (E) 180°. Note: Subplot (F) illustrates the definition of contact angle, and $U_0$ and $U$ stand for wind speed and inlet flow speed, respectively. Four mechanisms inducing liquid aerosol losses are described, including particle impaction on the outer wall after crossing the limiting streamlines (outside impaction), particle impaction on the inner wall, particle trapping by *vena contracta*, and inertial impaction at the bend.

due to stronger gravitational loss in the former case (*Hangal & Willeke, 1992*; *Okazaki, Wiener & Willeke, 1987b*; *Tufto & Willeke, 1982*). Based on a number of experiments on one thin-walled and five blunt samplers and two types of aerosols (five monodispersed aerosols with diameters of 5, 10, 21, 41, and 68 μm, respectively, and a polydispersed aerosol with a mass median diameter of 1.6 μm and a geometric standard deviation of 2.7), *Li & Lundgren (2002)* argued that a model proposed by Vincent group (*Tsai & Vincent, 1993*; *Tsai et al., 1995*; *Vincent, 1987*; *Vincent, 1989*) can be applicable to both thin-walled and thick-walled (blunt) probes (*Li & Lundgren, 2002*). This model describes the effect of wind contact angle θ on $\eta_{aspiration}$ for $0 \leq \theta \leq 90°$ as,

$$\eta_{aspiration} = \left[1 + \alpha_1 \left(\frac{S^2}{D_b^2 \varphi} \cos\theta - 1\right)\right]\left[1 + \alpha_2 \left(\frac{D^2}{S^2} - 1\right)\right] \tag{14}$$

$$\alpha_1 = 1 - \frac{1}{1 + g_1 r St \left(\cos\theta + 4\frac{S\sqrt{\sin\theta}}{D_b \sqrt{\varphi}}\right)} \tag{15}$$

$$\alpha_2 = 1 - \frac{1}{1 + g_2 St \varphi (D_b/S)^2}. \tag{16}$$

There are only a few studies on the inlet efficiency for contact angles beyond 90°. *Vincent et al. (1986)* evaluated the aspiration efficiency of a large diameter, thin-walled probe sampling aerosols with diameters ranging 6–34 μm and reported that $\eta_{aspiration}$ has smaller values for $\theta = 180°$ than $\theta \leq 90°$ due to the blunt leading edge of the probe body. Based on the method proposed by Vincent group, *Tsai & Vincent (1993)* derived two simple models for $\theta = 90°$ and $\theta = 180°$, respectively, for both thin-walled and blunt samplers as,

$$\eta_{aspiration} = \frac{1}{1 + 4 \times 2.21 St(r/\varphi)^{1/2}} \quad \text{for } \theta = 90° \tag{17}$$

$$\eta_{aspiration} = \frac{1}{1 + 4 \times 4.5 St \varphi^{1/3} r^{-0.29}} \quad \text{for } \theta = 180°. \tag{18}$$

However, for the $\theta = 90°$ case, *Brixey, Evans & Vincent (2005)* pointed out that the exponent of the $r/\varphi$ term for thin-walled samplers should be 1/4 instead of 1/2 based on the data collected under the experimental settings of $2.75 < R_u < 54.3$, and $0.0001 < St < 0$.

For transmission efficiency under non-isoaxial conditions, *Hangal & Willeke (1990b)*, by recognizing that the aerosol transmission loss occurs not only by aerosol gravitational settling, but also inertial impaction and *vena contracta* trapping at the inner wall (c.f. Figs. 1B–1D), proposed the following correlation for $0 \leq \theta \leq 90°$, $0.25 < R_u < 4$, and $0.02 < St < 4$,

$$\eta_{tg} = \exp[-4.7 K_\theta^{0.75}] \tag{19}$$

$$\eta_{ti} = \exp[-75(I_w + I_v)^2] \tag{20}$$

where $K_\theta = K \sqrt{\cos\theta}$, $I_w = St \sqrt{R_u} \sin(\theta - \alpha) \sin(0.5\theta - 0.5\alpha)$ for upward sampling, $I_w = St \sqrt{R_u} \sin(\theta + \alpha) \sin(0.5\theta + 0.5\alpha)$ for downward sampling, $I_v = 0.09[St \cos\theta(-1 + 1/R_u)]^{0.3}$ for $0.25 < R_u < 1$, $I_v = 0$ for $R_u \geq 1$, and $\alpha = 12[(1 - \theta/90) - \exp(-\theta)]$. Here, $I_w$ and $I_v$ account for the effects of direct impaction and *vena contracta*, respectively. *Sreenath, Ramachandran & Vincent (2001)* attempted to measure the inlet efficiency of a thin-walled sampler for aerosol particles with diameters ranging 1–37 μm and an angle range from 0 to 90° and generated a complex empirical correlation for $\eta_{transmission}$ with the applicable ranges of $0.38 < R_u < 3.8$ and $0.002 < St < 2$ as,

$$\eta_{transmission} = [1 - c_1 R_u^{c_2} St^{c_3}(1 - \exp(-c_4\theta))]\exp(-c_5 K_1^{c_6}) \tag{21}$$

where $c_1, c_2, c_3, c_4, c_5,$ and $c_6$ are empirical coefficients of 0.094, 1.799, 0.280, 142.78, 71.84, and 1.695, respectively, and $K_1 = 0.152 St(\cos\theta/U^{2.5} R_u)$. This model shows non-monotonic patterns with $St$ and $\theta$, indicating multiple mechanisms of transmission efficiency, including gravitational settling, direct impaction, and *vena contracta*. Later, they also adopted a similar approach to measure the inlet efficiency of a blunt (spherical) sampler for similar aerosol particles but a wider contact angle range from 0 to 180°

(*Sreenath, Ramachandran & Vincent, 2002*). They argued that both $\eta_{aspiration}$ and $\eta_{transmission}$ are independent of $\theta$ for backward sampling (*i.e.*, $\theta > 90°$), and $\eta_{aspiration}$ efficiency can be estimated using the model for $\theta = 180°$ proposed by *Tsai & Vincent (1993)*. However, as the $\eta_{aspiration}$ and $\eta_{transmission}$ at $\theta = 0$ were not directly measured, their arguments need further validation. Moreover, with Eqs. (19)–(21) only applicable for moving air, no transmission efficiency models are available for still air.

In summary, although a number of experimental studies have examined the effects of wind speed and contact angle on the aspiration and transmission efficiencies for different aerosols (c.f. Table 1), most research is focused on particles with a diameter greater than 1 µm and a limited wind contact angle range of $0 \leq \theta \leq 90°$. However, not much is known about sampling efficiency for submicron aerosols with $90° < \theta < 180°$. Considering that finer aerosols have more adverse effects on human health and the deployment of the inlet may not always have a low contact angle, in this study, we will attempt to fill this gap by providing a wind-tunnel-based dataset on sampling efficiency for a submicron liquid aerosol and a wider contact angle range of $0 \leq \theta \leq 180°$. The dataset will be used to evaluate the effect of wind speed and contact angle on sampling efficiency and to assess the applicability of classic models to submicron aerosols. Finally, the limitations and implications of this study will also be addressed.

## METHOD

The experiments were conducted in a suck-type mini wind tunnel composed of an intake, a working section, and an engine section, and deployed in a laboratory room where the air temperature was controlled at approximately 25 °C and the relative humidity at around 65% (Fig. 2). Air entered the wind tunnel through the intake and then proceeded to the working section, which has a dimension of 2.4 m in length, 0.3 m in width and 0.36 m in height. The engine comprises a diffusion segment and a voltage-controlled fan capable of adjusting the freestream wind speed.

As a low-cost and commonly used portable sampler, two units of TSI DustTrak 8530 were used to measure the aerosol concentration inside and outside the wind tunnel, respectively. Both DustTrak units remained outside the tunnel to avoid direct influence on the air flows within the working section. Each DustTrak pumped the air first through a 50 mm long metal inlet with its inner and outer diameters of five and seven mm, respectively, and then through a one-meter-long conductible tube with an inner diameter of five mm. Both tubes were kept as straight as possible to minimize aerosol losses in the tubing system and to ensure comparable transport efficiency between the two DustTrak sampling systems. The DustTrak has a self-regulating pumping system to control the flow rate, ensuring the internal flow speed in the metal inlet is at a constant of 2.55 m/s. With a known inner diameter of the tube, the Reynolds number $Re$ is around 850, indicating it is a laminar flow.

As our focus is on very fine particles, each tube was connected to the DustTrak inlets through a PM$_{2.5}$ impactor (TSI 854021) to filter out particles with diameters greater than 2.5 µm. As depicted in Fig. 2, one metal inlet was positioned inside the wind tunnel,

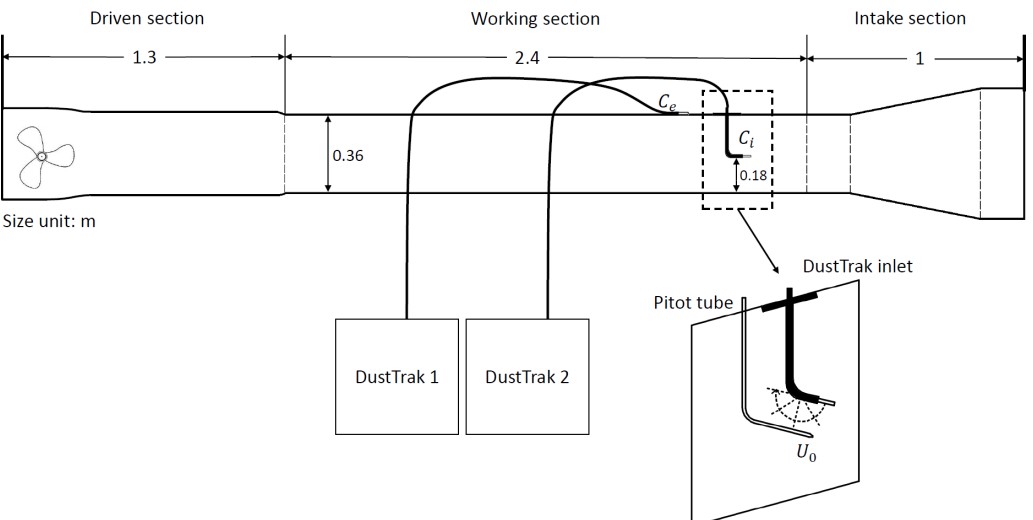

**Figure 2 Wind tunnel experimental setup.** Note: $C_e$ and $C_i$ are external and internal concentrations of the liquid aerosol, respectively, and $U_0$ is the freestream wind speed, with a range from 0.56 to 4.12 m/s.

situated 0.18 m above the bed and 0.15 m from the leading edge of the working section. The orientation of this metal inlet was adjusted using multiple 3D-printed plastic mounts to simulate various wind contact angles (c.f. Figs. 1F and 2). It is worth noting that the tube connecting the metal inlet sometimes has to be bent for inlet orientation. The bending angle is related to the wind contact angle $\theta$. For $\theta = 0$ and 180° (*i.e.,* inlet facing upwind and downwind, respectively), the bending angle is 90°. For $\theta = 90°$ (*i.e.,* inlet facing downward), the bending angle is 0. A pitot tube was installed spanwise to this metal inlet, positioned approximately 2 cm away, to measure the freestream wind speed. The other metal inlet was fixed horizontally outside the tunnel, away from either the intake or the engine, to prevent wind influence.

An Antari Z-350 Fazer was used to generate aerosol droplets from a mixture of water, triethylene glycol, and 1, 2-propylene glycol. Unlike solid aerosols, liquid particles are subject to substantial evaporation (*Lim et al., 2008*) but exhibit minimal resuspension after depositing on sampler walls (*Wang & John, 1988*). The instrument was manually set to its minimum continuous output to produce a stable fog. However, this manual adjustment led to some variability in the resulting particle mass concentration. To characterize the aerosol, we used two instruments to measure the particle size distribution: a water-based Condensation Particle Counter (TSI CPC 3789) for droplets smaller than 421 nm and an Optical Particle Sizer (TSI OPS 3330) for droplets larger than 334 nm. To characterize the stabilized aerosol (as described below), each instrument continuously collected six two-minute samples of the particle size distribution. Unsurprisingly, substantial evaporation of aerosol was observed right after the fog was generated. However, an Optical Particle Sizer measurement showed that the mass distribution for particles larger than 300 nm became stabilized after 30 min, with very small fluctuations (less than ±10%) in mass

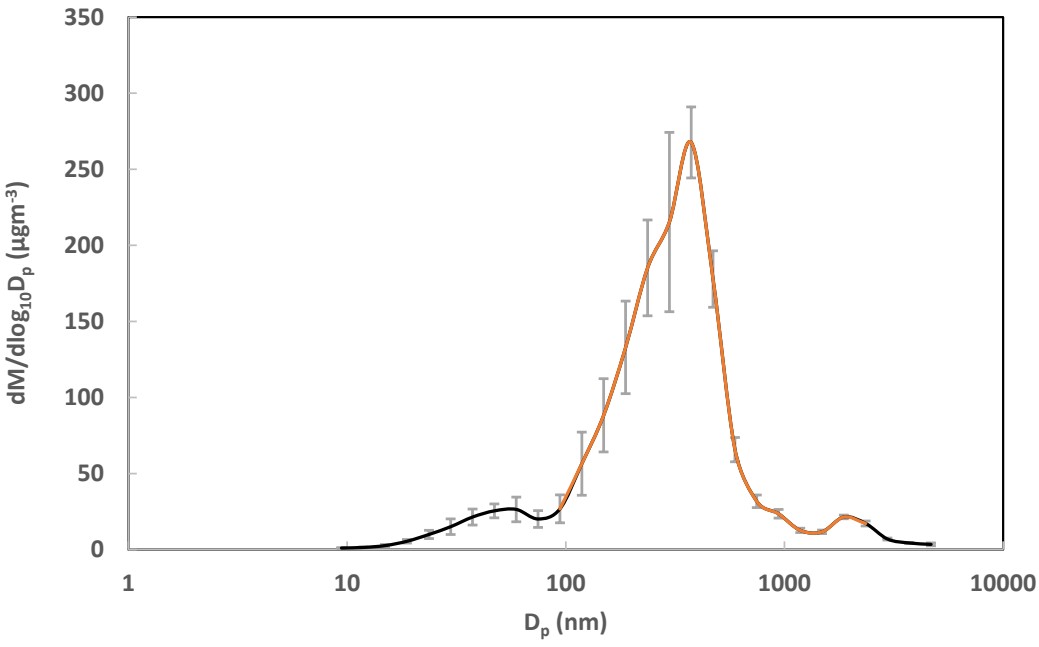

**Figure 3 Mass-based aerosol particle size distribution.** Note: The orange line indicates the portion of aerosols that were actually measured by DustTrak units. Error bars indicate the standard deviation of the measurements. $D_p$ represents the particle diameter, and $\mathrm{d}M/\mathrm{dlog_{10}}D_p$ stands for the mass concentration per logarithmic size interval.

distribution (c.f. Supplemental Information 3). Therefore, the fog producer was running for over 30 min before the mass distribution measurements commenced.

The data from the two instruments were then processed and merged. First, using the known aerosol density ($1.04 \times 10^3$ kg/m$^3$) and assuming spherical particles, the count distribution from each sample was converted to a mass distribution *via* the TSI Aerosol Instrument Manager software (v. 9.0). Because the instruments have different size bins, we standardized the data by defining a new set of bins with a uniform logarithmic interval of 0.1 and proportionally allocating the mass from the original bins (see particlesizedistribution.xlsx in the supplementary materials for details). A correction was then applied to account for concentration shifts between the non-simultaneous CPC and OPS measurements. This was done by calculating a correction factor from the ratio of their average mass concentrations at a common bin midpoint (374.5 nm). Finally, the average mass distribution from the CPC was merged with the corrected average distribution from the OPS. The resulting full mass distribution (10.58 nm to 4,712 nm) was unimodal, with a mass median diameter of 0.285 μm and a geometric standard deviation of 1.95 (Fig. 3). The standard deviation for each new bin was calculated using the same correction factor, presented as error bars in Fig. 3. Since the DustTrak units cannot measure aerosols smaller than 0.1 μm, the measured portion is depicted as the orange curve in Fig. 3, occupying 89% of the mass distribution.
The experiments involved testing five different contact angles (0°, 45°, 90°, 135°, and 180°, c.f. Fig. 2), with three replicate experiments initially performed for each angle. For each experiment, the fog producer and wind tunnel engine ran for at least half an hour before data collection to ensure the aerosol was well mixed both inside and outside the wind tunnel, and then inter-calibration was conducted by co-locating the two metal inlets in still air. After inter-calibration, the difference in PM mass concentration measurements was less than 1 $\mu g/m^3$, compared to the aerosol concentrations during the experiments, which were in the order of $10^2$ to $10^3$ $\mu g/m^3$. Each experiment comprised 13 runs at controlled wind speeds with varying wind speeds up to 4.1 m/s. During each run, the DustTrak and pitot tube recorded measurements for six minutes at a sampling frequency of 5 Hz under constant wind speed conditions. To maintain data quality, experiments where aerosol concentrations fell outside the DustTrak optimal operation range ($10^2$–$10^3$ $\mu g/m^3$) were excluded. This resulted in a total of 12 valid experiments: two replicates each for 0°, 90°, and 180°, and three replicates each for 45° and 135°. Additionally, for each experiment, runs with wind speeds below the pitot tube's operable range (<0.5 m/s) were discarded. Ultimately, 144 clean runs with a wind speed range of 0.56–4.12 m/s were retained for further analysis.

Following each run, average values for internal and external PM mass concentrations ($C_i$ and $C_e$, respectively) and the freestream wind speed ($U_0$) were obtained. A sampling efficiency term $\eta$ was then derived after referencing the concentration measured under windless conditions, *i.e.*,

$$\eta = C_i/C_e. \tag{22}$$

Note that, as sampling efficiency for still air cannot reach 100%, $\eta$ can only be considered as a parameter to quantify the effect of wind speed and contact angle on sampling efficiency. Subsequently, with the efficiency values derived under different wind speed and contact angle conditions, regression analysis was employed using the *fitlm* function provided by MATLAB 2022a (MathWorks Inc., Natick, MA, USA) to evaluate their individual and combined effects and to develop an empirical model for $\eta$ predictions.

To evaluate the individual effects of wind speed and contact angle, the 144-sample dataset was systematically partitioned. First, to analyze the influence of wind speed, the dataset was divided into five subsets by contact angle ($\theta$), resulting in sample sizes of 24, 34, 24, 36, and 25 for $\theta = 0°, 45°, 90°, 135°,$ and 180°, respectively. For each subset, regression analysis was conducted to examine the relationship between sampling efficiency $\eta$ and wind to inlet flow speed ratio $R_u$. Second, to assess the effect of contact angle, the dataset was re-partitioned into eight subsets based on half-meter-per-second wind speed intervals (from 0.5 to 4.5 m/s), resulting in sample sizes of these subsets ranging from nine to 27. This approach minimized the confounding influence of wind speed variation. Similarly, for each subset, regression analysis was also conducted to examine the relationships between $\eta$ and $\cos\theta$. For both regression analyses, the regression strategy involved initially testing linear models and progressing to higher-order polynomials if the fit was insufficient, with non-significant terms being systematically eliminated from the final form of the equation. Finally, based on the regression equations from individual effect analysis, a general model

for the combined effect was formulated. After deriving the general model's coefficients by fitting it to the entire 144-sample dataset, its predictive accuracy was evaluated by linearly regressing predicted $\eta$ values against observed $\eta$ values with the intercept set to 0. For all models, the goodness of fit was evaluated using the coefficient of determination ($R^2$) and $p$ values.

To make comparisons with previous works, the sampling efficiency predicted from the empirical model in this study ($\eta$) was compared with that calculated from previous models ($\eta_o$) using the same particle size (we chose 1 μm here as a conservative criterion), wind speed, and contact angle ranges. Here, $\eta_o$ can be defined as,

$$\eta_o = \frac{\eta_{aspiration}\eta_{transmission}\eta_{transport}}{\eta'_{aspiration}\eta'_{transmission}\eta'_{transport}} \tag{23}$$

where $\eta'_{aspiration}$ and $\eta'_{transmission}$ are the aspiration and transmission efficiencies for the sampler with its inlet deployed horizontally outside the tunnel, and $\eta_{transport}$ and $\eta'_{transport}$ are transport efficiencies for tubes deployed inside and outside the wind tunnel, respectively.

Here, as the inlet of the metal inlet has a wall thickness of 1 mm and does not meet thin-wall criteria (*Davies, 1968*), aspiration efficiency $\eta_{aspiration}$ was derived using the Vincent group model for cylindrical thick-walled samplers (Eq. (14)). As the air movement outside the wind tunnel is minimal, $\eta'_{aspiration}$ was determined using the aspiration efficiency model for the calm air (Eq. (9)).

With a similar tubing system, aerosol loss in these two samplers should be the same, except for the bend for simulating various contact angles inside the wind tunnel. Therefore,

$$\frac{\eta_{transport}}{\eta'_{transport}} = \frac{\eta_{bend}}{1} = \eta_{bend}. \tag{24}$$

Here, the magnitude of $\eta_{bend}$ was evaluated using a model for submicron particles with diameters ranging from 0.18 to 0.63 μm in a bend of 90° (*Sato, Chen & Pui, 2003*; *Tsai & Pui, 1990*), where aerosol loss at the bend is the greatest:

$$\eta_{bend,90} = 1/(1 + St_m^{f_1 Stk_m + f_2}) \tag{25}$$

where $f_1 = -0.2435$ $f_2 = -2.574$, and $St_m = St_t/(1 + A)\exp(-0.001)De - B\sqrt{De}$. Here, the Stokes number for tube $St_t = D_p^2 c \rho_p U/(18\mu D)$, Dean number $De = ReD^{0.5}R_b^{-0.5}$, $A = 2.193 + 0.06055DR_b^{-1} + 0.4096D^{0.5}R_b^{-0.5}$, $B = 0.08946 + 0.02981D^{-0.5}R_b^{0.5}$, and $R_b$ is the centerline radius of curvature of the bend, varying from 2 to 5 times of tube inner diameter when bend angle was at 90°. After calculation, $\eta_{bend,90}$ has a minimum of 0.9999999975, indicating the aerosol loss at the bend can be neglected because $\eta_{bend}$ should be higher than $\eta_{bend,90}$ when the bend angle is smaller than 90°.

According to Eq. (10), $\eta_{transmission}$ was indirectly derived using $\eta_{tg}$ and $\eta_{ti}$ values from Eq. (19) and 20 for blunt samplers. However, the evaluation of $\eta'_{transmission}$ is an issue as there are no available transmission efficiency models for still air. With a fixed flow rate, $\eta'_{transmission}$ should be a constant for the whole experiment. Here, we adopted $\eta'_{transmission} = 1$ as an approximation. With Eq. (24) and negligible aerosol loss at the bend, Eq. (23) can be

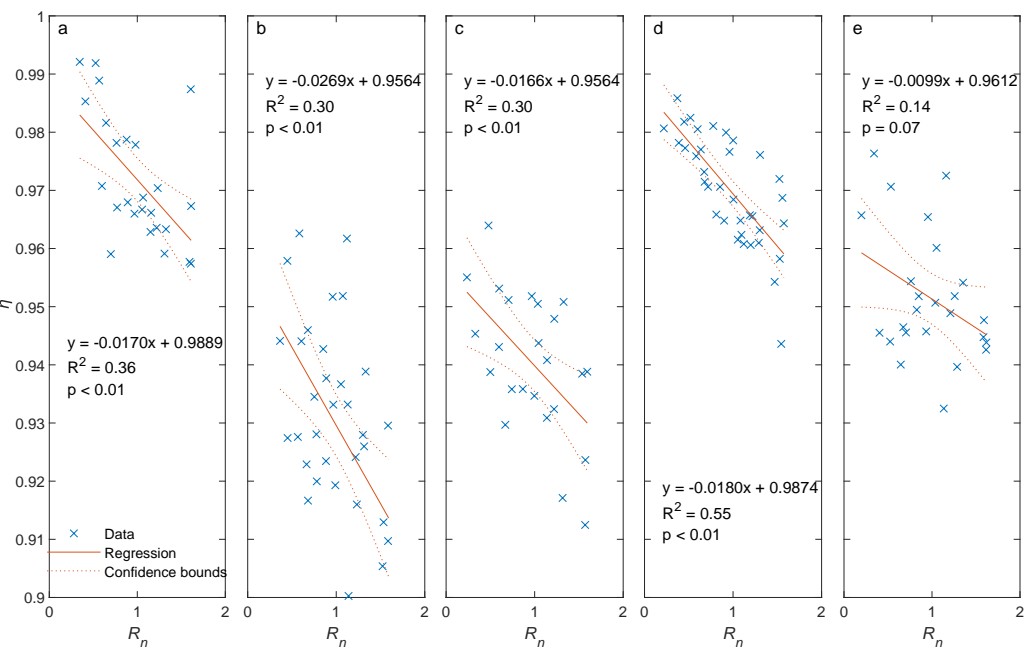

**Figure 4** Sampling efficiency ($\eta$) as a function of the ratio of wind speed to inlet flow speed ($R_u$) for different contact angles ($\theta$): (A) 0°, (B) 45°, (C) 90°, (D) 135°, and (E) 180°. Note: Blue cross symbols represent the observed data, solid red lines represent the regression lines, and red dashed lines are the 95% confidence bounds. The regression equations, $R^2$ values, and $p$ values are also displayed.

rewritten as,

$$\eta_o = \frac{\eta_{aspiration}\eta_{tg}\eta_{ti}}{\eta'_{aspiration}} \tag{26}$$

which was used to derive $\eta_o$. Note that, given the limited application range of $\eta_{tg}$ and $\eta_{ti}$, only the $\eta$ and $\eta_o$ results for $0 \leq \theta \leq 90°$ are compared.

# RESULTS

## Individual effect

As shown in Fig. 4, sampling efficiency $\eta$ is generally high, with all $\eta$ values larger than 0.9. A regression analysis reveals that $\eta$ has a negative linear relationship with normalized wind speed $R_u$ for each contact angle $\theta$ (Fig. 4). All $R^2$ values are equal to or greater than 0.3 and $p$ values are smaller than 0.01, except for $\theta = 180°$ ($R^2 = 0.14$ and $p = 0.07$). The regression functions have intercept values ranging from 0.9564 to 0.9889 and slope values from $-0.0269$ to $-0.0099$.

As described in the methods, the relationship between sampling efficiency ($\eta$) and contact angle ($\theta$) was evaluated using eight data subsets grouped by wind speed. A cubic polynomial model was found to best describe this relationship. However, to ensure statistical confidence, the analysis presented here is limited to the six subsets with a sample size of at least 15. The two subsets corresponding to the wind speed intervals of 0.5–1.0 m/s and 3.5–4.0 m/s were excluded due to their small sample size of 9. As shown in Fig. 5, a

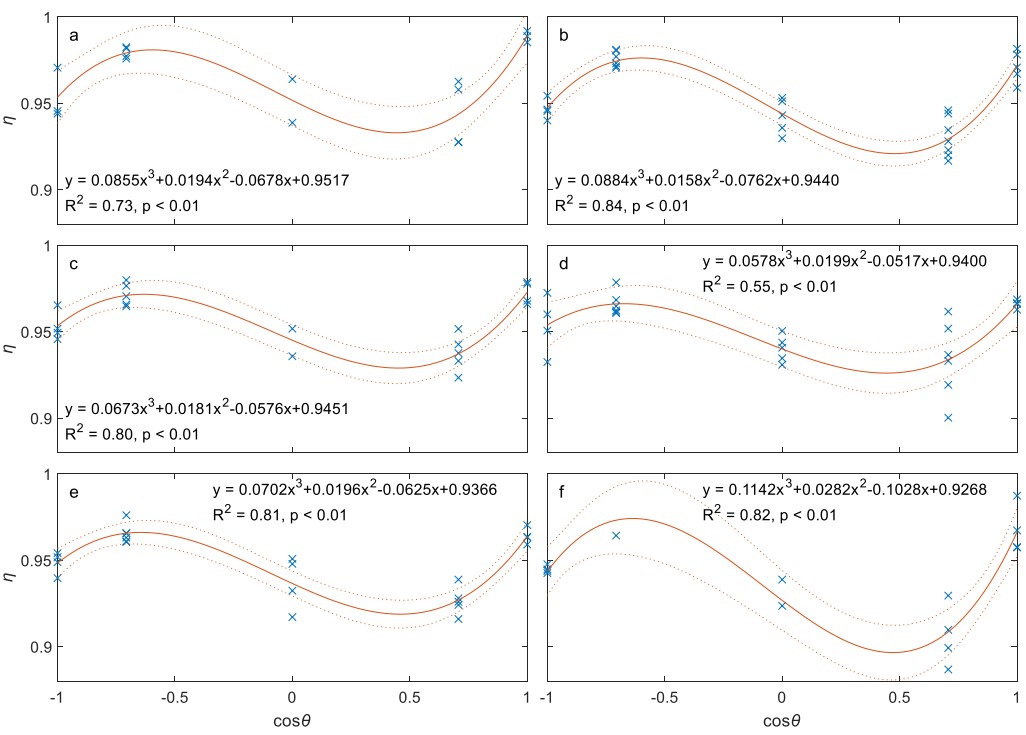

**Figure 5** **Sampling efficiency ($\eta$) as a function of contact angles ($\theta$) for different wind speeds ($U_0$): (A) 1–1.5 m/s, (B) 1.5-2 m/s, (C) 2–2.5 m/s, (D) 2.5-3 m/s, (E) 3–3.5 m/s, and (F) 4.0–4.5 m/s.** Note: Blue cross symbols represent the observed data, solid red lines represent the regression lines, and red dashed lines are the 95% confidence bounds. The regression equations, $R^2$ values, and $p$ values are also displayed.

statistically significant cubic polynomial relationship exists between $\eta$ and $\cos\theta$ for all 6 wind speed cases, with two peaks at $\theta \approx 0°$ and 135° and one valley at $\theta \approx 60°$, indicating that the effect of $\theta$ is complex. All $R^2$ values are greater than 0.55, and $p$ values are smaller than 0.01.

## Combined effect

Regression analysis was also applied to all data. Given $\eta$ has a significant or marginally significant ($p < 0.1$) relationship between $\eta$ and $R_u$ and a significant cubic polynomial relationship ($p < 0.01$) with $\cos\theta$, the following relationship can be used to describe how $R_u$ and $\cos\theta$ affect sampling efficiency $\eta$:

$$\eta = a\cos^3\theta + bR_u\cos^3\theta + c\cos^2\theta + dR_u\cos^2\theta + e\cos\theta + fR_u\cos\theta + gR_u + h \qquad (27)$$

where a–h are empirical coefficients. After removing the terms with $p$-values larger than 0.05, the following model is derived with $p < 0.01$ for all coefficients and $R^2 = 0.751$:

$$\eta = 0.07710\cos^3\theta + 0.02208\cos^2\theta - 0.06672\cos\theta - 0.01788R_u + 0.95685. \qquad (28)$$

There are no interaction terms in Eq. (28), indicating wind speed and contact angle appear to affect the sampling efficiency separately and independently. The modeled $\eta$ values, along with observed data, are depicted in Fig. 6. For the same wind speed, an increase in contact

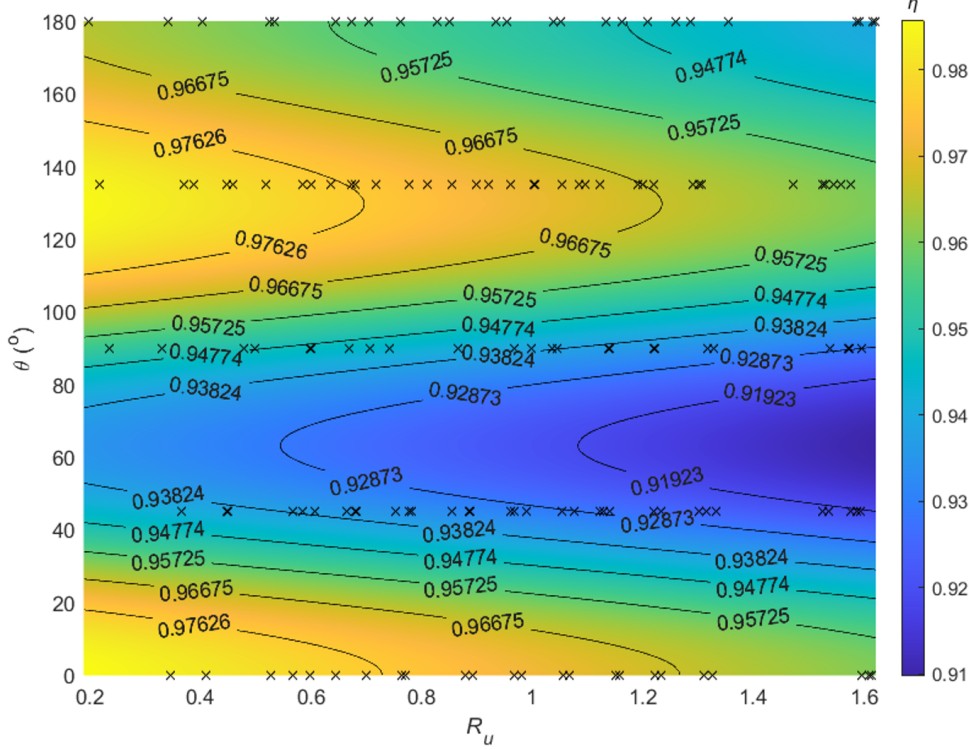

**Figure 6 Sampling efficiency ($\eta$) as a function of the ratio of wind speed to inlet flow speed ($R_u$) and contact angle ($\theta$).** Note: The black cross symbols show the raw data points, while the labeled curves are isolines representing constant sampling efficiency values.

angle from 0° to 180° (or from 1 to −1 for $\cos\theta$) causes a rise-fall-rise pattern in $\eta$. For the same contact angle, a larger $R_u$ reduces the efficiency. High $\eta$ values are located at $\theta \approx 0°$ and 135° with low wind speed, and low $\eta$ values are located at $\theta \approx 45°$ with high wind speed. Regression analysis demonstrates good consistency between predicted $\eta$ and observed $\eta$, with $R^2 = 0.67$, $p < 0.01$, and $RMSE = 0.011$ (c.f. Fig. 7).

## Comparison with classic models

Following the procedure described in the method section, $\eta_{aspiration}$, $\eta_{tg}$, $\eta_{ti}$, and $\eta'_{aspiration}$ were calculated using classic models and plotted against contact angle $\theta$ for multiple $R_u$ values of 0.5, 1, and 2 (Fig. 8). Aerosol loss from aspiration in still air $(1 - \eta'_{aspiration})$ and gravitational settling $(1 - \eta_{tg})$ is one order of magnitude lower than that from aspiration in the moving air $(1 - \eta_{aspiration})$. Conversely, $\eta_{ti}$ has a similar magnitude as $\eta_{aspiration}$ when $R_u < 1$. With increasing wind speed, both $\eta_{aspiration}$ and $\eta_{ti}$ increase, but $\eta_{tg}$ decreases. With increasing contact angle, $\eta_{tg}$ and $\eta_{ti}$ (when $R_u < 1$) also increase, while $\eta_{aspiration}$ and $\eta_{ti}$ (when $R_u < 1$) vary very little.

With known $\eta_{aspiration}$, $\eta_{tg}$, $\eta_{ti}$, and $\eta'_{aspiration}$, $\eta_o$ can be derived using Eq. (26) for given $\theta$ and $R_u$ values (0° < $\theta$ < 90° and $R_u = 0.5$, 1, and 2). Similarly, $\eta$ can also be calculated using the model derived from this study (Eq. (28)) for the same values of $\theta$ and $R_u$. As shown in Fig. 9, all $\eta_o$ values are significantly higher than $\eta$ values over 0° < $\theta$ < 90°. With

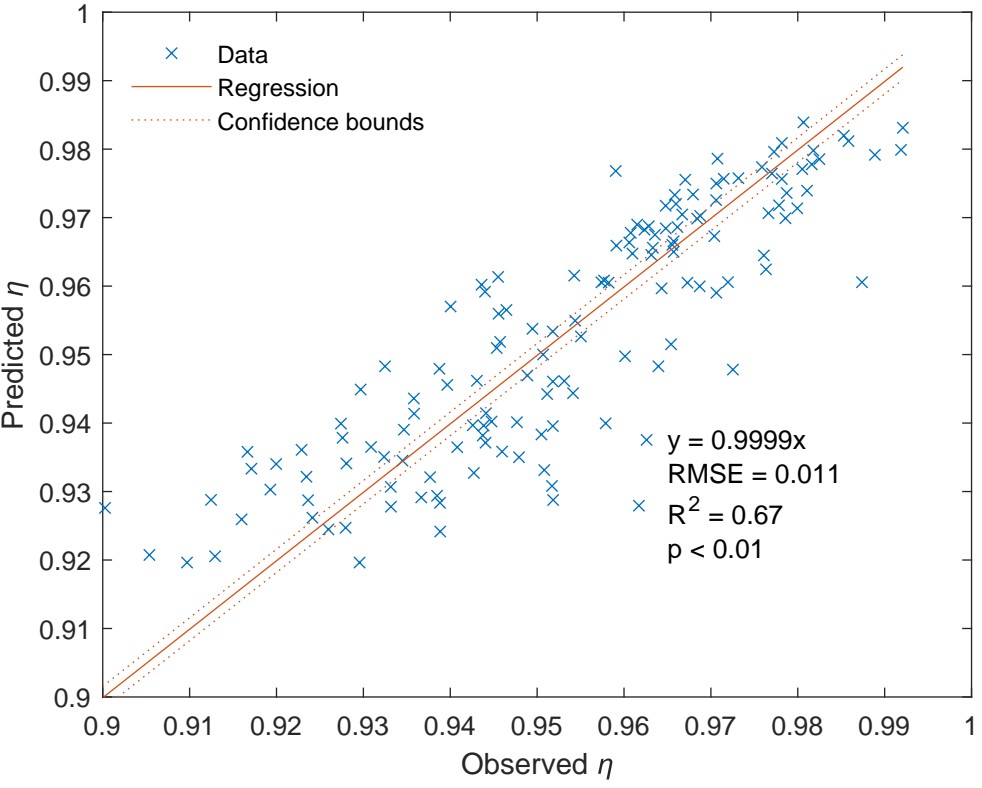

**Figure 7 Predicted sampling efficiency (denoted as Predicted $\eta$) vs. observed sampling efficiency (denoted as Observed $\eta$).** Note: Blue cross symbols represent the data, solid red lines represent the regression lines, and red dashed lines are the 95% confidence bounds. The regression equation, together with $R^2$, RMSE, and $p$ values, is also displayed.

the increase of $\theta$, $\eta_o$ increases while $\eta$ first decreases and then increases. Conversely, with increasing $R_u$, $\eta_o$ increases while $\eta$ decreases.

## DISCUSSIONS

### Interpretations of the results

We have generated an empirical model to evaluate the effect of wind speed and contact angle on the sampling efficiency for a type of submicron liquid aerosol using a sampling system with a customized thick-walled cylindrical metal inlet connected to a TSI DustTrak 8,530 unit using a conductive tube. Our results show that the sampling efficiency $\eta$ is positively correlated to wind speed or $R_u$, and $\eta$ also follows a rise-fall-rise pattern with contact angle $\theta$ varying from 0 to 180°. These findings are substantially different from classic models, where the sampling efficiency $\eta_o$ generally rises with increasing $\theta$ or decreasing $R_u$ (c.f. Fig. 9). To explain this discrepancy, the aerosol loss/gain pathways must be re-examined. For our case, aerosol loss/gain can be mainly attributed to five processes: (1) aerosol deviation from streamlines outside the inlet, (2) aerosol gravitational settling, (3) the existence of *vena*

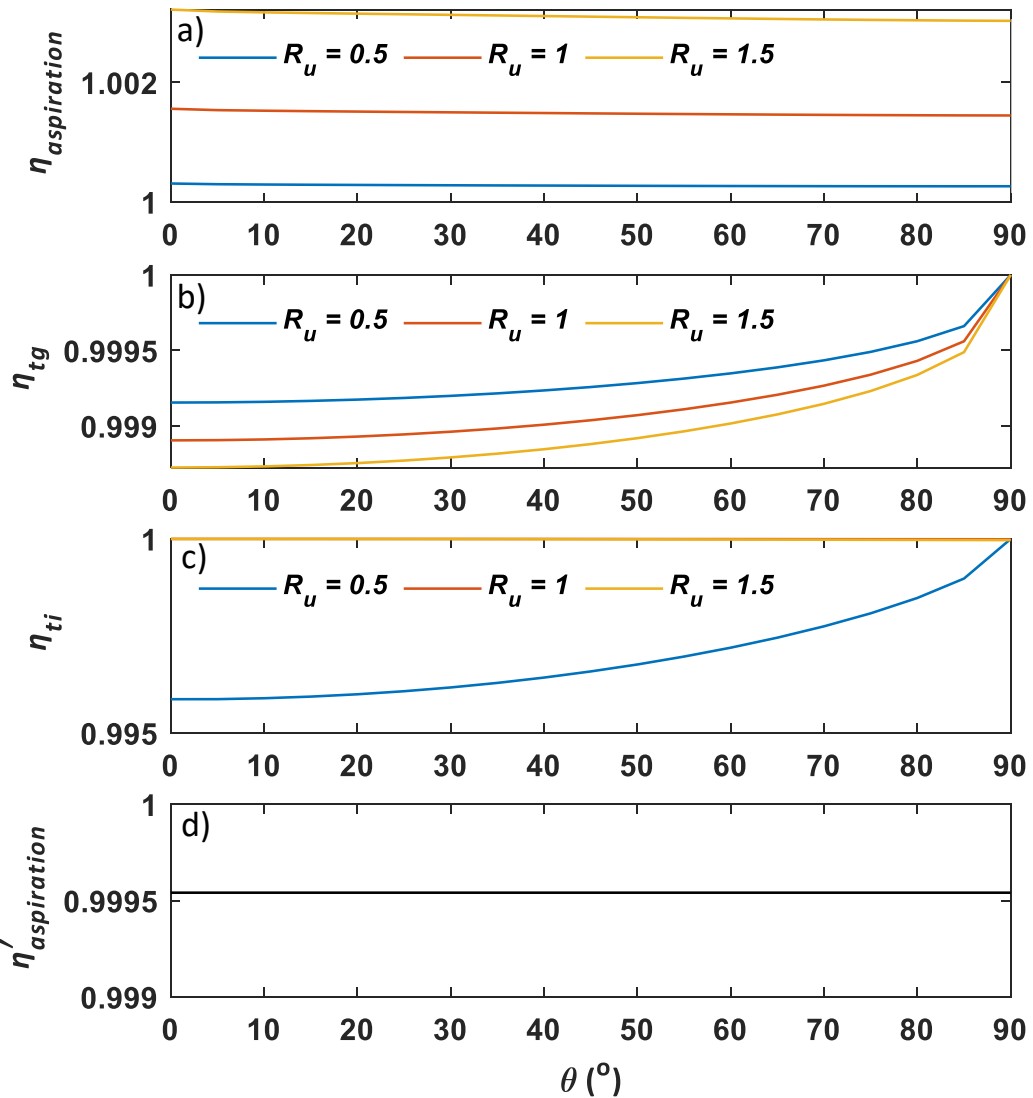

**Figure 8** Sampling efficiency components, including (A) $\eta_{aspiration}$, (B) $\eta_{tg}$, (C) $\eta_{ti}$, and (D) $\eta'_{aspiration}$, as a function of $\theta$ and $R_u$. Here, $\eta_{aspiration}$, $\eta_{tg}$, $\eta_{ti}$, and $\eta'_{aspiration}$ are aspiration efficiency, gravitational transmission efficiency, inertial transmission efficiency, and aspiration efficiency in the calm air, respectively, and $\theta$ and $R_u$ stand for contact angle and wind to inlet flow speed ratio, respectively.

*contracta*, (4) inertial particle impaction at the inlet inner wall near the entrance, and (5) inertial particle impaction at the tube bend.

The first process is related to the curvature of the streamlines. Under super-isokinetic conditions when $R_u < 1$, the limiting streamlines are bent outwards (c.f. Fig. 1A), some larger particles inside may escape from the limiting streamline and impact the outer wall or front edge of the inlet due to larger inertial and gravitational effects, leading to aerosol loss (*Davies & Subari, 1982*; *Grinshpun, Willeke & Kalatoor, 1993*; *Hangal & Willeke, 1990b*; *Lipatov et al., 1988*). Conversely, under sub-isokinetic conditions when $R_u > 1$, some aerosol particles outside the streamlines may surpass them and enter the inlet because

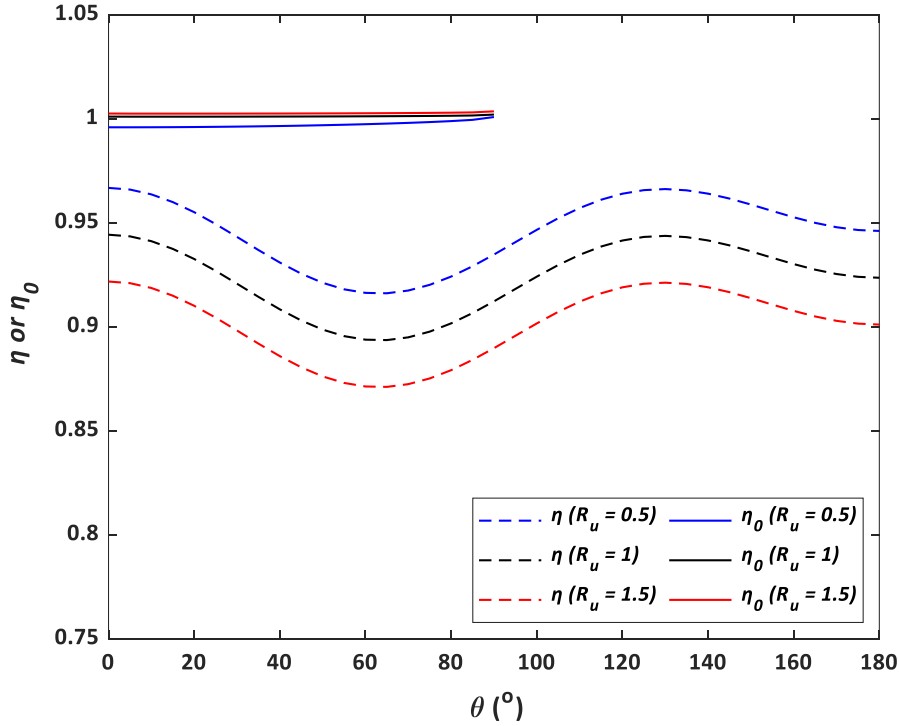

**Figure 9** Predicted sampling efficiencies from Eq. (28) ($\eta$) and Eq. (26) ($\eta_0$) as a function of contact angle ($\theta$) and wind to inlet internal speed ratio ($R_u$).

the streamlines are bent inwards, resulting in aerosol gain (*Belyaev & Levin, 1972*; *Belyaev & Levin, 1974*; *Hangal & Willeke, 1990b*; *Liu, Zhang & Kuehn, 1989*; *Okazaki, Wiener & Willeke, 1987a*; *Wiener, Okazaki & Willeke, 1988*). This effect was quantitatively examined by modelling aspiration efficiency for thick-walled cylindrical inlets using Eq. (5) (*Tsai & Vincent, 1993*; *Tsai et al., 1995*; *Vincent, 1987*; *Vincent, 1989*) (c.f. $\eta_{aspiration}$ curves in Fig. 8). However, the aspiration efficiency for liquid aerosols may be overestimated by the Vincent group's model, as their validation typically used solid aerosols for which particle rebound could lead to re-entry into the inlet. Therefore, although *Li & Lundgren (2002)* showed that Vincent's model can be valid for particles as small as 1.6 $\mu$m, its applicability to our case is still difficult to determine without direct measurement of aspiration efficiency.

The second process is related to the aerosol gravitational settling into the boundary layer of the inner wall during aerosol transmission in the metal inlet. As described in Eq. (19), gravitational transmission efficiency, $\eta_{tg}$, is usually quantitatively evaluated using a model proposed by *Hangal & Willeke (1990b)* based on a number of empirical studies on liquid particles with their smallest diameter of 2.5 $\mu$m (*Okazaki, Wiener & Willeke, 1987a*; *Okazaki, Wiener & Willeke, 1987b*; *Tufto & Willeke, 1982*). These studies have the Stokes numbers as low as 0.01, which, however, is still much larger than ours ($1.2 \times 10^{-4}$ to $9.8 \times 10^{-4}$ by assuming the particle size of 1 $\mu$m). Therefore, the scalability of Hangal & Willeke's model in $\eta_{tg}$ estimation in our case is questionable.

The third and fourth processes are related to inertial transmission efficiency $\eta_{ti}$. According to Eq. (20) (*Hangal & Willeke, 1990b*), the contributions from *vena contracta* and inertial impaction can be represented by $\exp[-75(I_v)^2]$ and $\exp[-75(I_w)^2]$, respectively. For better illustration, these two terms are respectively plotted against contact angles from 0 to 90° for four $R_u$ values of 0.25, 0.5, 1, and 2 in Fig. 10. As *vena contracta* only exists when $R_u < 1$ (*McCabe, Smith & Harriott, 1993*), $\exp[-75(I_v)^2] = 1$ for $R_u \geq 1$. Aerosol loss increases with decreasing wind speed due to the increasing magnitude of eddies. Two eddies can be formed when $\theta = 0°$ (c.f. Fig. 1A), and they gradually diminish until an angle of 90°, which explains why there is an increasing trend of $\exp[-75(I_v)^2]$. Conversely, aerosols can also directly deposit on the inner wall near the entrance and at the bend of the plastic mounts due to inertial impaction. Stronger wind tends to have greater impaction loss, which can explain the negative relationship between $\exp[-75(I_v)^2]$ and $R_u$. As for contact angle, as aerosol impaction is minimal when streamlines are parallel to the wall, it is not surprising to see that $\exp[-75(I_v)^2]$ reaches the minimum at $\theta = 90°$ (c.f. Fig. 1C) and maximum at $\theta = 0°$. As shown in Fig. 10, the magnitude of aerosol loss due to direct impaction is much smaller than that from *vena contracta* for $R_u < 1$, which makes $\eta_{ti}$ mainly controlled by the existence and magnitude of *vena contracta* under super-isokinetic conditions (c.f. $\eta_{ti}$ curve in Fig. 8). However, this conclusion is based on experimental data with a limited range of Stokes number as indicated above; therefore, its validity requires further examination.

The last process is related to the inertial loss of aerosol particles at the tube bend, which is smallest for a straight tube with $\theta = 90°$ (c.f. Fig. 1C), and the largest for $\theta = 0°$ (c.f. Fig. 1A). However, this effect was not discussed here as the aerosol loss due to tube bending is in the order of $10^{-9}$ (c.f. method section), much smaller than that from other processes.

Since the sampling efficiency for the external sampler is constant, the pattern of $\eta_0$ should be primarily controlled by $\eta_{aspiration}$, $\eta_{tg}$, $\exp[-75(I_v)^2]$, and $\exp[-75(I_w)^2]$. As shown in Figs. 8 and 10, the total sampling efficiency from our experiment, $\eta$, has demonstrated a similar pattern with $\exp[-75(I_w)^2]$, but not with $\exp[-75(I_v)^2]$, indicating that the effect of inertial particle impaction at the inner wall may be substantially underestimated and/or the impact of *vena contracta* may be substantially overestimated by *Hangal & Willeke (1990b)*. However, as mentioned above, previous models are based on experimental data for particles larger than 2.5 µm, therefore, further empirical exploration on submicron aerosols is needed to fully explain this discrepancy.

## Error analysis

The liquid aerosol used in this study presented a potential for error, as evaporation could decrease particle size over time. To address this, we allowed the aerosol to stabilize for at least 30 min (well above 30 min in most cases) after initial mixing before beginning data collection. A pilot study confirmed that after this period, the aerosol reached equilibrium, with no significant changes observed in the mass distribution for particles larger than 300 nm (c.f. Supplemental Information 3). While the mass distribution for particles smaller than 300 nm was not measured, we expect the effect of evaporation to be minimal after 35 min, as larger particles are more susceptible to deposition.

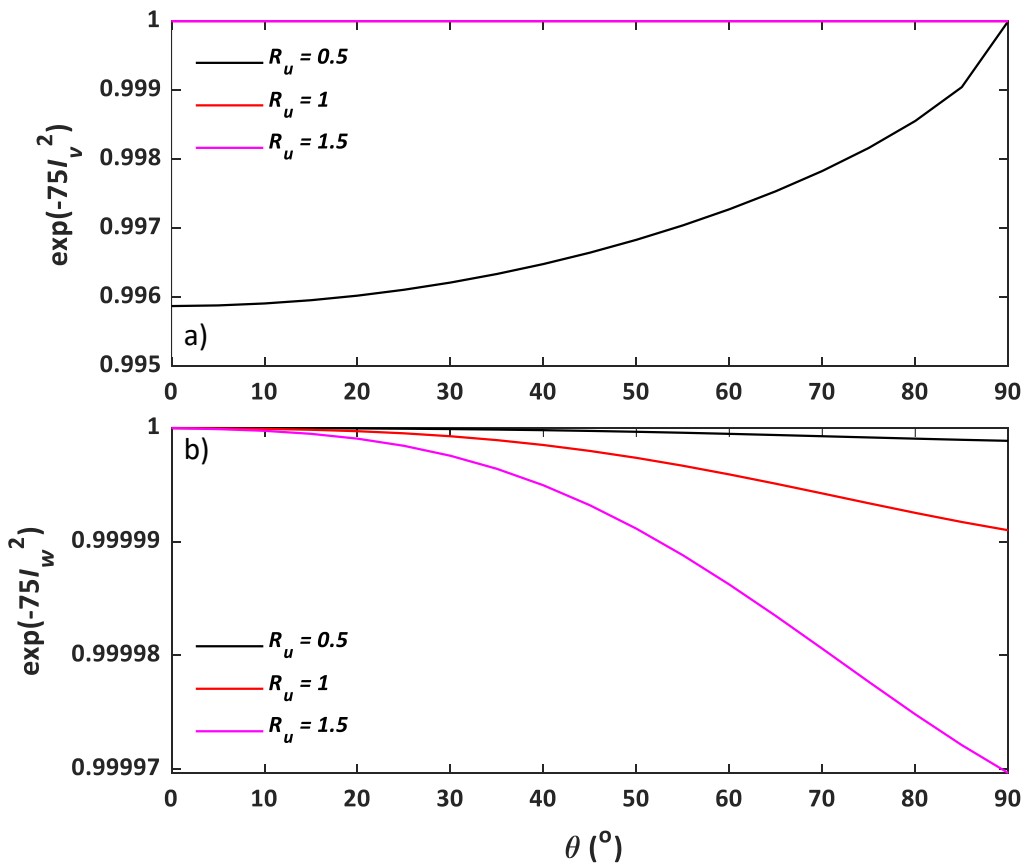

**Figure 10** **Transmission efficiency components accounting for (A) aerosol loss from *vena contracta*** ($\exp(-75I_v^2)$) **and (B) inertial impaction** ($\exp(-75I_w^2)$) **as a function of $\theta$ and $R_u$.** Here, $\theta$ is the contact angle and $R_u$ is the ratio of wind speed to inlet flow speed.

The wide distribution of particle size and lack of mass distribution after sampling make it difficult to identify which size should be used for $\eta_0$ calculation. As mentioned above, we expect the aerosol deposition to be primarily controlled by the inertial particle impaction on the inner wall. Therefore, we chose a relatively large particle size of 1 μm to calculate $\eta_0$ values because larger particles have higher inertia and more aerosol loss. This arrangement would inevitably underestimate $\eta_0$; however, choosing a smaller size can only result in a more significant discrepancy between $\eta$ and $\eta_0$, which still generally supports our main findings.

It is also worth noting that we have attempted to minimize the experimental errors by careful intercalibration between the two samplers before each experiment. During intercalibration, the two metal inlets were collocated outside the wind tunnel to measure the aerosol concentrations under windless conditions. The samplers were tuned until the relative difference between the two readings was equal to or smaller than 0.001 mg/m³. With our aerosol concentration varying from 0.2 to 0.9 mg/m³, the instrument error ranged between 0.1% and 0.5%, generally much smaller than the level of the measured aerosol loss

(0.5–9%). We expect this instrument error to occur randomly, which can partially explain why the $R^2$ values for some regressions are not very high.

## LIMITATIONS

There are some limitations of our study. First, we defined a relative sampling efficiency term $\eta$ (or $\eta_0$), as the ratio of mass concentration under wind conditions to that under windless conditions. Although they do not represent the absolute sampling efficiency, with a fixed sampling efficiency under windless conditions, the variability of them can be used to assess the effects of wind speed and contact angle.

Second, the empirical equation proposed here is only applicable to a specific type of liquid aerosol and a specific sampling system used in this study. Considering that solid particles can rebound and the rebounding characteristics vary substantially with different impacting speeds and angles (*Wang & John, 1988*), solid aerosols should have higher efficiency than liquid aerosols (*Koehler et al., 2012*). This study adopted a common sampling instrument but with a customized thick-wall cylindrical inlet, and this can result in substantially different flow response and aerosol loss patterns near the metal inlet compared with thin-wall or other types of blunt inlets (*Li & Lundgren, 2002*). Additional experimental evidence for solid or mixed-phase aerosols and various inlets is needed to fully understand the mechanisms of aerosol deposition.

Third, our model's scalability is limited by the experimental parameters. Although we tested a wide range of contact angles, the sampling inlet was rotated only in a vertical plane, primarily facing downward. Furthermore, the model was derived from a restricted wind speed range (up to 4.1 m/s). Consequently, its applicability to other scenarios—such as horizontal rotation, upward-facing inlets, or wind speeds beyond this range—remains uncertain and requires further investigation.

## IMPLICATIONS

Sampling inlets used for air quality monitoring in environments such as wind tunnels, pipes, or chambers are typically unshielded and directly exposed to the flow. While prior research often assumed that losses of submicron particles were negligible (*Chung & Dunn-Rankin, 1992*; *Grinshpun, Willeke & Kalatoor, 1993*; *Liu, Zhang & Kuehn, 1989*; *Tufto & Willeke, 2014*), our study reveals these losses can be as high as 9%, leading to a significant underestimation of particulate matter (PM) levels. Given that accurate aerosol measurement is essential for early warnings of hazardous air quality (*Luo et al., 2021*; *Qiu et al., 2022*), this work has substantial implications for protecting public health.

This study indicates that total sampling efficiency is highest at the lowest wind speed when the sampler is positioned at a 0° or 135° angle relative to the wind. Consequently, we recommend these as the optimal operating conditions for sampler users. This conclusion diverges from the established recommendation to employ both isoaxial and isokinetic sampling to obtain representative aerosol samples (*Brockmann, 2011*). While our findings affirm the high efficiency of an isoaxial orientation (*i.e.,* contact angle of 0), they argue against the need for isokinetic conditions. We consistently found that super-isokinetic

sampling—where the inlet flow speed is greater than the wind speed—results in a higher sampling efficiency. Therefore, the re-evaluation of aerosol measurement protocols is needed.

While acknowledging the limitations of our study (*e.g.*, vertical inlet rotation and a constrained wind speed range), the discrepancy between $\eta$ and $\eta_o$ highlights an opportunity to expand upon conventional theories for submicron aerosols. The results suggest that the relative contributions of inertial impaction and *vena contracta* trapping may be different from previously predicted, which could influence classic models' applicability to submicron particle transmission. Considering the unverified scalability of aspiration models, this finding underscores the need for careful consideration when applying classic theories beyond their validated range. In addition, this study provides a valuable dataset across a wide spectrum of contact angles (0–180°), offering a resource for aerosol scientists to assess and further develop predictive models for submicron aerosol efficiency.

The proposed sampling efficiency model has significant potential for field applications in both ambient air monitoring and occupational exposure assessment. For ambient monitoring, the model can be used to develop correction factors for portable samplers, improving data accuracy in the face of variable wind speeds and orientations. In occupational settings, where airflows are often more predictable, the model can be integrated into sampling protocols to more precisely assess worker exposure to hazardous aerosols. By enhancing the reliability of measurements in both scenarios, this work can help facilitate more timely public health warnings to mitigate potential exposure to hazardous aerosols (*Luo et al., 2021*; *Qiu et al., 2022*).

## CONCLUSIONS

Through wind tunnel experiments on a widely used monitoring system, we evaluated a liquid-based aerosol under various wind speeds and contact angles. Our results show that sampling efficiency decreases as wind speed increases and exhibits a complex, non-monotonic relationship with the wind contact angle (0–180°). Contrary to classic models that suggest wind has a negligible effect on the sampling efficiency of submicron particles for unshielded inlets, this study also demonstrates significant wind-induced aerosol losses as high as 9%, representing a significant margin of error for air quality monitoring. This study provides a valuable dataset for validating future sampling efficiency models, and the empirical models proposed herein can be used to correct measurements made with similar sampling systems.

This substantial discrepancy between our model and classic models reveals the limited applicability of conventional theories to submicron aerosols and highlights potential flaws in current aerosol measurement protocols and air quality warning systems. Future investigations should determine how key parameters—including inlet geometry, wind speed, contact angle, and sampling plane (horizontal *vs.* vertical)—affect the sampling efficiencies of submicron aerosols in both solid and liquid phases.

## ACKNOWLEDGEMENTS

Many thanks to Dr. Defeng Zhao and Yali Jin from Fudan University for assisting the aerosol size distribution measurement. During the preparation of this work the authors used AI tools, such as Grammarly and ChatGPT, to improve the readability and language of the manuscript. After using these tools, the authors reviewed and edited the content as needed and take full responsibility for the content of the published article.

### Funding

This study was financially supported by a Research Development Fund (No. RDF-17-01-31) from Xi'an Jiaotong-Liverpool University. The funders had no role in study design, data collection and analysis, decision to publish, or preparation of the manuscript.

### Grant Disclosures

The following grant information was disclosed by the authors:
Research Development Fund, Xi'an Jiaotong-Liverpool University: RDF-17-01-31.

### Competing Interests

The authors declare there are no competing interests.

### Author Contributions

- Bokun Sun conceived and designed the experiments, performed the experiments, analyzed the data, prepared figures and/or tables, authored or reviewed drafts of the article, and approved the final draft.
- Ziyang Wang performed the experiments, authored or reviewed drafts of the article, and approved the final draft.
- Jiayun Huang performed the experiments, prepared figures and/or tables, and approved the final draft.
- Yumeng Li performed the experiments, prepared figures and/or tables, and approved the final draft.
- James R Cooper conceived and designed the experiments, authored or reviewed drafts of the article, and approved the final draft.
- Lei Han conceived and designed the experiments, authored or reviewed drafts of the article, and approved the final draft.
- Bailiang Li conceived and designed the experiments, authored or reviewed drafts of the article, and approved the final draft.

### Data Availability

The raw measurements are available in the Supplementary Files.

## Supplemental Information

Supplemental information for this article can be found online at http://dx.doi.org/10.7717/peerj.20235#supplemental-information.

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
