# Peer review of "Control of wind speed and contact angle on submicron particulate matter sampling"

_PeerJ, doi:10.7717/peerj.20235_

## Round 0.1 · original submission · Major Revisions

· Academic Editor

Major Revisions

I wish to thank you for this submission. The reviewers revised your manuscript, and they raised important problems. Please, read carefully the referees' suggestions and follow them to improve the manuscript quality.

Reviewer 1 ·

Basic reporting

The abstract lacks sufficient depth in presenting the core findings and their implications. It should include more quantitative results and a clearer statement of the significance of the study.

The introduction provides a broad context but does not clearly articulate the research gap and the novelty of the study. The literature review should be expanded to include recent studies that highlight the unresolved issues in the field.

The hypotheses and objectives of the study need to be explicitly stated. The current framing is somewhat implicit, making it difficult to discern the main research questions being addressed.

Experimental design

The methodology section lacks clarity in experimental design and data collection procedures. Precise details on sample size determination, replication strategy, and statistical assumptions should be provided to enhance reproducibility.
The description of analytical techniques needs improvement. It is essential to specify software versions, parameters used in statistical tests, and any modifications made to standard procedures.

Validity of the findings

The results section is descriptive but lacks critical analysis. More in-depth interpretation of the findings with reference to existing literature would strengthen the discussion.

The figures and tables are informative but require better labeling and captions. Some figures are too complex and should be simplified for better readability. Ensure that all abbreviations used in tables and figures are explained.

The discussion does not sufficiently integrate the study’s findings with prior research. A comparative analysis with recent studies would provide better context and validate the study's contributions.

The limitations of the study are not adequately discussed. A dedicated subsection should outline potential biases, limitations in methodology, and directions for future research.

The conclusion is somewhat generic and does not provide a strong summary of key findings. It should reinforce the significance of the results and their implications for further research or practical applications.

There are several minor grammatical and typographical errors throughout the manuscript. A thorough proofreading is necessary to enhance readability and clarity.

Citations and references should be carefully checked for completeness and correctness. Ensure all references follow the journal’s prescribed format and include recent relevant studies.

Ethical considerations and conflict of interest statements should be explicitly mentioned if applicable, ensuring compliance with journal requirements.

Additional comments

The manuscript would benefit from a clearer narrative flow, particularly in transitioning between sections. Improving logical coherence would enhance the overall readability of the paper.

The practical implications of the findings are not well elaborated. Expanding on how the results can be applied in real-world scenarios would strengthen the impact of the study.

A graphical abstract or conceptual diagram summarizing the study’s key findings could enhance reader engagement and understanding.

Overall, the manuscript presents an interesting study, but it requires significant revisions in clarity, methodological transparency, and depth of discussion to meet the standards for publication.

·

Basic reporting

(1) The manuscript is well-written. However, the abstract is technical and crowded, especially for an interdisciplinary readership. Terms like “cosine of wind contact angle expressed as a third-order polynomial” should be simplified to convey the meaning without sacrificing accuracy. Consider a concluding sentence that highlights how users can apply this method.
(2) Figures, although data-rich, are often cluttered (e.g., Figs. 5, 6, 9). Captions should be expanded to help interpret the content independently from the main text. Axes must include units and more descriptive labels.
(3) The literature review provides relevant context, but more emphasis should be given to recent advancements in PM₁ monitoring, especially for portable devices.

Experimental design

(1) The absence of post-sampling particle size distribution data poses a limitation in understanding deposition mechanisms. Since aerosol loss is inherently size-dependent, interpretations related to vena contracta or inertial impaction remain somewhat speculative. Including size-distribution data post-sampling, or at least acknowledging this limitation more directly, would add valuable depth to the study.
(2) The study focuses on a single liquid aerosol type without exploring other particle forms such as solids or mixed-phase aerosols. Given the known differences in sampling behavior between these types, a brief discussion or, if feasible, experimental comparison would help clarify how broadly the findings may apply.
(3) Although the study includes triplicate experiments, most figures do not present standard deviations or confidence intervals. Including error bars and relevant statistical analyses (e.g., ANOVA or Tukey’s HSD) would considerably enhance the perceived robustness and reproducibility of the trends described.
(4) The claim that the aerosol was “well mixed” following 30 minutes of equilibration is reasonable, but no supporting data (such as spatial concentration profiles) are provided. A tracer-based test or concentration uniformity check particularly under varying wind conditions would help substantiate this key assumption.

Validity of the findings

(1) The conceptual explanation of aerosol loss mechanisms (e.g., vena contracta, inertial impaction, streamline curvature) is insightful but remains theoretical. The study would be greatly strengthened by supporting these hypotheses with direct experimental validation or computational fluid dynamics (CFD) simulations.
(2) The effect of particle size and phase is not fully explored, particularly since only liquid aerosols were used. The findings may not directly apply to solid particles, which behave differently upon impaction due to rebound characteristics. This should be more clearly acknowledged as a limitation.

Additional comments

(1)The study fills an important knowledge gap, particularly for PM1 sampling under varied wind conditions.
(2) Future versions of the manuscript would benefit from:
(a) Adding CFD or flow visualization data to validate mechanisms.
(b) Extending the experimental design to include solid particles.
(c) Providing a practical tool or correction framework for field users.
(3) The authors are encouraged to revise the abstract for greater clarity and impact, summarizing key findings in less technical terms.
(4) Overall, the work has strong potential but requires careful revision to enhance its scientific rigor and practical relevance.

---

## Round 0.2 · Minor Revisions

· Academic Editor

Minor Revisions

Before the final acceptance of your paper, please follow the suggestions of the reviewer #2 for increasing the manuscript quality.

Reviewer 1 ·

Basic reporting

Sufficiently improved

Experimental design

Significantly Revised

Validity of the findings

Sufficiently improved

Additional comments

As the authors have incorporated all my suggestions and addressed all comments, I would recommend this article for its potential publication in this current format. Thanks

·

Basic reporting

The authors have made a substantial effort to improve the manuscript. The abstract has been enriched with quantitative results, and the introduction has been expanded with more comprehensive literature review. However, manuscript could be improved by addressing following comments.
1. A few equations and symbols appear distorted due to formatting issues; these should be carefully corrected for readability.
2. Figure captions could be expanded to better describe experimental conditions (e.g., wind speed range, aerosol type).
3. Ensure consistent use of terms such as “submicron” vs “very fine” particles throughout the text.
4. References are adequate, but a stronger connection to very recent literature (2023–2025) on PM₁ measurement technologies would enhance relevance.

Experimental design

The use of a wind tunnel and controlled aerosol source is appropriate, and replication is mentioned. However, addressing following comments would enhance the manuscript.
1. Clarify the number of replicates and provide more details on statistical treatment of replicate data.
2. Expand the explanation of aerosol characterization, particularly regarding the stability of particle size distribution over time.
3. While liquid aerosols are a reasonable surrogate, acknowledge more explicitly the potential differences from real-world solid particle behavior.

Validity of the findings

The findings are presented systematically, and the empirical model derived for sampling efficiency is useful. The authors appropriately compare their results to classic models and highlight discrepancies.
1. Add more details on statistical evaluation (e.g., include confidence intervals or standard errors for regression models).
2. Some strong claims in the discussion (e.g., “inapplicability of classic models”) should be moderated by noting experimental limitations.
3. The discussion of limitations could explicitly mention the restricted range of wind speeds tested (0.9–4.5 m/s).

Additional comments

The manuscript is substantially improved from its earlier version and makes a useful contribution. With the minor revisions above, it will be suitable for publication.
1. Include a short paragraph on potential field applications (ambient air monitoring vs occupational exposure).
2. Ensure consistent notation in equations and figures.

---

## Round 0.3 · accepted · Accept

· Academic Editor

Accept

The authors addressed all the referees' comments and now the paper can be published in PeerJ.